# Sixth International Conference on Duckweed Research and Applications Presents Lemnaceae as a Model Plant System in the Genomics and Postgenomics Era

**DOI:** 10.3390/plants12112134

**Published:** 2023-05-28

**Authors:** Viktor Oláh, Klaus-Juergen Appenroth, Eric Lam, K. Sowjanya Sree

**Affiliations:** 1Department of Botany, Faculty of Science and Technology, University of Debrecen, 4032 Debrecen, Hungary; olahviktor@unideb.hu; 2Matthias Schleiden Institute—Plant Physiology, University of Jena, 07743 Jena, Germany; klaus.appenroth@uni-jena.de; 3Department Plant Biology, Rutgers State University of New Jersey, New Brunswick, NJ 08901, USA; 4Department of Environmental Science, Central University of Kerala, Periye 671320, India

**Keywords:** biomass production, duckweed, Lemnaceae, whole genome sequencing

## Abstract

The 6th International Conference on Duckweed Research and Applications (6th ICDRA) was organized at the Institute of Plant Genetics and Crop Plant Research (IPK) located in Gatersleben, Germany, from 29 May to 1 June 2022. The growing community of duckweed research and application specialists was noted with participants from 21 different countries including an increased share of newly integrated young researchers. The four-day conference focused on diverse aspects of basic and applied research together with practical applications of these tiny aquatic plants that could have an enormous potential for biomass production.

## 1. Introduction

Duckweeds are aquatic monocots that exhibit a high degree of miniaturization and simplification of their plant body (Figure 1). By efforts from the community of researchers and application specialists, these plants have been re-established as model plants in the current genomics and postgenomics era [1]. Over this course, the first International Conference on Duckweed Research and Applications (ICDRA) was organized in 2011 at the Chinese Academy of Science in Chengdu, China [2]. A crucial step in the evolution of this community was the establishment of the International Steering Committee on Duckweed Research and Applications (ISCDRA) in 2013 at the second ICDRA organized at the Rutgers University, New Brunswick, NJ, USA [3]. Steered by the ISCDRA, the biennial meetings continued to be organized at the Kyoto University, Japan (2015), the Central University of Kerala, India (2017), and the Weizmann Institute of Science at Rehovot, Israel, (2019), discussing and deliberating on the advances made in the field over each two-year interval. Most recently, the 6th ICDRA was organized on behalf of the ISCDRA (Figure 2) at the Institute of Plant Genetics and Crop Plant Research (IPK) in Gatersleben, Germany, from 29 May to 1 June 2022, which has a one-year delay because of the COVID-19 pandemic. Ingo Schubert from the IPK served as the Chair, and Manuela Nagel, also of the IPK, and Klaus-J. Appenroth, University of Jena, Germany, were the Co-Chairs (Figure 3). A special issue “Duckweeds: Research meets applications” was organized by three guest editors for the journal *Plants* (Basel) (Figure 4).

The four-day conference included forty-two oral presentations and thirty-one posters, most of them presented by young researchers or Ph.D. students, with a total of ninety-four participants from twenty-one different countries. The increasing number of research groups entering the duckweed research field is an important development for our community as this brings in expertise from varied fields to work on duckweeds.

Apart from the lecture and poster sessions, a public lecture was also delivered by Klaus-J. Appenroth on the topic “A new crop plant with great potential for nutrition, water treatment and energy”, in view of the potential use of duckweeds in various practical applications. This session was intended for and attended by the public in and around the IPK as outreach to the society at large. As customary for the final day of the conference, a General Assembly of the participants voted and decided for the next conference to be held in 2024 at Bangkok, Thailand. This will be organized by Arinthip Thamchaipenet from Kasetsart University, Thailand, and Metha Meetam of Mahidol University, Thailand.

The following sections on (1) genomics, evolution and genome organization; (2) differentiation and diversity; (3) physiology, metabolism and microbiome; (4) stress, toxicology, cryopreservation; (5) wastewater remediation; (6) large-scale cultivation; and (7) feed and food, will highlight recent advances in the diverse areas of duckweed research and applications.

## 2. Genomics, Evolution and Genome Organization

The report about “Genomic and epigenomic consequences of clonal growth habit in the Lemnaceae” (Rob Martienssen, Cold Spring Harbor, NY, USA), the first “Invited talk” of the conference, provided evidence that the genomes of Lemnaceae selectively lost genes required for RNA-directed DNA methylation that are involved in transposon silencing. Further, triploid hybrids that have arose from *Lemna minor* and *Lemna turionifera* are commonly found. He also reported that they presumably form through hybridization with unreduced gametes, and that divergent “ZMM class” mismatch repair genes could support polyploid meiosis. Several intraspecific hybrids were analyzed by the method of **T**ubulin-**B**ased **P**olymorphism (TBP). Species described by the late Elias Landolt as *Lemna japonica* turned out to be hybrids between *L. minor* and *L. turionifera*. In some cases, also hybrids between *Lemna gibba* and *L. minor* were uncovered (“TBP fingerprinting unveiled interspecific hybridization in the genus *Lemna*”, Laura Morello, Milano, Italy). Transcriptomic data revealed the absence and non-expression of key components including the RNA-directed DNA methylation (RdDM) pathway methyltransferase (DRM2), as well as the lack of siRNA associated with transposable elements (“Epigenetic regulation of transposable elements in duckweeds”, by Rodolphe Dombey, Gregor Mendel Institute, Austria). Some preliminary results from ongoing long-term experimental evolution studies in mesocosms that aim to investigate plant evolution in multitrophic communities were reported by Shuqing Xu (University of Mainz, Germany). In the report via virtual zoom link on “Duckweed genome architecture” it was reported that whole genome sequence data of at least one species from all genera of Lemnaceae are now available, very recently supplemented by that of *Landoltia punctata*, *Wolffia australiana* and *Wolffiella neotropica* (Todd P. Michael, Salk Institute for Biological Studies, USA). Whereas *Spirodela intermedia* and *Spirodela polyrhiza* share a similar genome architecture [4], *Wolffia* and *Lemna* revealed dynamic evolutionary trajectories, whole genome duplication events and polyploidization. For the first time, a survey on genome size measurement for all 36 duckweed species, on chromosome counts for 31 species and on their evolutionary impact was performed using flow cytometry and cytogenetic approaches (Phuong Thi Nhu Hoang, Dalat University, Vietnam). While chromosome numbers for all 36 species has been published [5], only half of them are based on recent, more reliable measurements. Soon progress in this field should be expected from additional advances in high-throughput single molecule long-read sequencing. Due to the high copy number of ribosomal DNA in higher plants, conserved coding sequence and more rapidly evolving spacer sequences, the rDNA has become a favorite subject for studies related to plant systematics, evolution and biodiversity. Investigating rDNA in a large number of duckweed species, the Borisjuk group (Nikolai Borisjuk, Huaiyin Normal University, China) concluded that these put duckweeds in the spotlight for research on the molecular evolution of the rDNA.

## 3. Differentiation and Diversity

The series of oral presentations started with a highlight as a plenary talk about “Duckweed hibernation: unravelling the molecular basis of the turion induction switch in *Spirodela polyrhiza*” (Eric Lam, Rutgers University, NJ, USA). “Hibernation” means the formation of hibernacles, better known as turions. Using resources with a newly assembled genome for *S. polyrhiza* 9512, which has a high turion yield coupled with a fast turion formation rate, the transcriptome of mature turions (survival organs) were compared with that of normal vegetative fronds. This was enabled by a new method developed for isolation of high quality RNA from turions despite their high content of starch and tannins. Identification and informatics analysis of global transcripts with changes of more than eight-fold between turions and fronds, it was found that genes in the pathways for stress responses, dormancy and in several biosynthetic pathways to increase starch and lipid synthesis are likely to play a special role in turion biology [6]. Especially convincing was the comparison of the results of *S. polyrhiza* to those of *S. intermedia*, a species with very similar properties [4,7] but unable to produce turions, where many of the turion-specific genes are no longer induced by the low phosphate trigger in spite of the presence of their orthologs in the genome. Finally, the first evidence for epigenetic changes in the transition between frond and turion tissues were also presented from a global analysis of cytosine methylation patterns in these two states.

Applying light and confocal microscopy with high levels of spatial resolution, the process of proliferating daughter fronds in *Wolffia globosa*, *W. australiana* and *L. gibba* was investigated (Ljudmilla Borisjuk, Leibnitz-Institute Gatersleben, Germany). This work showed that budding of new meristems started from very early stage onwards during formation of the nascent daughter fronds. Metabolome analysis (LC/MS) and spatially resolved infrared imaging (FTIR) in five genotypes of *Wolffia* with distinct growth characteristics were carried out. The integrative analysis of structural organization, meristematic activity and metabolism is expected to provide a better understanding of growth dynamics in duckweed. The life cycle of *S. polyrhiza* has been subjected to numerous studies that have revealed several key processes involved in turion formation and function. As reviewed and concluded by Paul Ziegler (University of Bayreuth, Germany), these processes might be a model for other turion-producing aquaphytes [8]. The developmental switch from frond propagation to turion formation is not primarily due to the decreasing day lengths and temperatures, but rather to nutrient depletion (e.g., phosphate) of the water habitat. The newly formed turions are dormant and require a prolonged exposure to cold water to be able to germinate, after which the plant’s early development (germination and growth of the newly formed fronds) will be fueled by two distinct carbohydrate reserves, sugars and starch. Development of roots in duckweeds was investigated in face of the fact that species in two genera (*Wolffiella*, *Wolffia*) do not have roots at all. The authors (Alexander Ware, University of Nottingham, UK) could show by careful comparative cytological studies that roots in duckweeds are vestigial as suggested already on a morphological basis by Gorham in 1941 [9]. Different clones of *Wolffiella hyalina* have different photoperiodic requirements for flowering induction and respond differently to exogenous application of Salicylic acid or Benzoic acid (Minako Isoda, Kyoto University, Japan). These results suggest natural variation of floral inductive pathways between accessions in *W. hyalina*. Moreover, flowering of non-flowering *W. hyalina* was induced when co-cultured with flowering *W. hyalina* even in the absence of SA or BA in the medium. This result suggests the existence of plant-to-plant communication for floral induction. Duckweed sexual reproduction is still one of the most unknown aspects in this plant family. For the first time, transcriptomic analysis was conducted on duckweed pistils, anthers, seeds and fronds from the short-day plant *Lemna aequinoctialis* (“Unravelling the genetic mechanisms of plant sexual reproduction in duckweed”, Cristian Mateo-Elizalde, Cold Spring Harbor, USA). These results should improve our understanding of the molecular pathways involved in vegetative and generative propagation in duckweeds.

A basic requirement to investigate biodiversity in the field of duckweeds is the reliable identification of species and clones. A significant progress in the genus *Lemna* was made possible by the application of the method of tubulin-based polymorphism [10]. Except for the species *S. polyrhiza* [11], it is not yet known how this method will perform in other genera (Laura Morello, IBBA CNR, Milano, Italy). Anton Stepanenko and co-worker (Huaiyin Normal University, China) investigated the species diversity in some regions of Ukraine and Eastern China. Only by using molecular methods, i.e., a two-barcode protocol with the chloroplast atpH–atpF and psbK–psbI spacer sequences as described previously [12], six species from Ukraine and six species from China could be identified reliably. *Lemna aequinoctialis* does not form a uniform taxon and therefore, the phylogenetic status of this species requires further investigations.

## 4. Physiology, Metabolism and Microbiome

Duckweeds have long earned their position as models in plant physiology research, and a series of presentations gave diverse examples of the current state-of-the-art research topics that were addressed with these tiny plants. In his invited talk, Tokitaka Oyama (Kyoto University, Japan) presented a study on the circadian regulation in duckweeds using transient reporter-gene expression-based bioluminescence monitoring. The evolutionary and physiological significance of such circadian traits and regulation has been less explored so far, but potentially has high relevance in plant adaptive mechanisms. The presented system allows analyses at frond- and cell-specific levels to track synchronization or uncoupling between cells in the circadian regulation amongst and within duckweed fronds [13]. They found the stability of circadian cycle to be genotype-dependent: *Lemna* species performed more stable rhythms, while those in the *Wolffiella* genus showed arrhythmia under constant irradiation and high temperature [14]. Similarly, intraspecific populations of *L. aequinoctialis* collected in diverse latitudes showed differences in critical daylengths and circadian periods, thus suggesting microevolutionary adaptation of circadian traits to local environmental factors [15].

Wisuwat Songnuan (Mahidol University, Thailand) focused on characterizing the potential of two duckweed species from Thailand to sequester CO_2_ on the basis of their high growth rates. Moshe T. Halpern (ARO, Volcani Institute, Israel) depicted another exciting field of application for duckweed models. One of the current concerns in agriculture is understanding the physiological background of declining protein content in crops under elevated ambient CO_2_ levels [16]. This can be caused by both decreasing nitrogen uptake and restricted nitrate reduction, and duckweeds, especially species of the *Wolffia* genus, may facilitate resolving this urgent problem. *Wolffia* offers an ideal system, in which the lack of roots excludes any possible influence of the root-to-shoot transport and leaves the shoot as the only organ for N-uptake and NO_3_^−^ assimilation. In addition, duckweeds may not only tolerate but prefer NH_4_^+^ as a nitrogen source and can also be cultured under aseptic conditions, thus limiting interference by such factors as nitrification normally taking place in soil-based systems. Speaking of nitrogen metabolism, revealing mechanisms of uptake and assimilation of nitrogen by duckweeds is of high practical significance on its own. Yet, despite its importance for wastewater remediation and biomass production, this field needs to be developed further. Olena Kishchenko (Huaiyin Normal University, China) presented their latest results on the topic with six duckweed species, shedding light on the complex regulation of nitrogen assimilation genes and confirming that these plants, though can easily utilize both nitrogen sources, prefer NH_4_^+^ to NO_3_^−^ [17].

The aquatic lifestyle, simple anatomy and ability to thrive under axenic conditions also makes duckweeds well suited to study plant–microbe interactions [18]. Three presentations addressed this topic from various aspects. An invited talk by Asaph Aharoni introduced ongoing research at Weizmann Institute of Science (Israel) with *L. minor* to identify metabolic pathways and key metabolites that regulate the colonization and composition of microbiome community through plant exudates. What does a microbiome look like, however, when there are no roots? Osnat Gillor’s presentation (Ben Gurion University of the Negev, Israel) focused on this question by studying the rootless *W. globosa*. It was hypothesized that the microbiome of this unique plant may be a composite of functionally distinct communities that normally inhabit separate parts of the plant body. The metagenomic analyses revealed a bacteria-dominated microbiome with a characteristic endophyte community. This community was not only different from that of the surrounding medium but also from that of epiphytes, and besides aiding phosphorus and iron uptake, it was likely capable of fixing CO_2_ and atmospheric nitrogen, as well as synthesizing vitamin B12. Besides friendly microbes, however, there are always hostile ones surrounding the plants. Dynamics and interactions of two duckweed-associated bacteria strains (*Bacillus* sp. MRB10 and *Chryseobacterium* sp. 27AL) with the cyanobacterium (also known as blue-green algae) *Microcystis aeruginosa* and duckweed *L. gibba* suggest that proper growth of host plants requires an appropriate population mixture and size, as it was described by Masaaki Morikawa (Hokkaido University, Japan). They found that even though duckweed-associated bacteria could support growth of plants while modulating that of *M. aeruginosa*, they could also become deleterious to the host plants when they were suspended instead of being attached to the plant surface.

## 5. Stress, Toxicology, Cryopreservation

As it was mentioned earlier, many presentations in the conference discussed turions of *S. polyrhiza* from various aspects. In her invited talk, K. Sowjanya Sree (Central University of Kerala, India) reported turion formation in another duckweed species for the first time. The enigmatic species *Wolffia microscopica* has recently been re-discovered [19] and, so far, was the only *Wolffia* species that had no known turion formation in the genus. Besides turion formation, K. Sowjanya Sree discussed another typical duckweed response to adverse conditions, which is starch accumulation. She presented that, while phosphate and nitrogen starvation can induce rapid carbohydrate accumulation in all genera of the Lemnaceae family, sulphate limitation had only marginal effects in that regard [20].

Besides stressors of natural origin, anthropogenic factors can also threaten duckweeds. Philippe Juneau (University of Quebec in Montreal, Canada) reported that even if aminomethylphosphonic acid (AMPA), which is one of the primary degradation products of the herbicide glyphosate, did not disturb growth and photosynthesis of duckweeds at environmentally-relevant concentrations, it nevertheless decreased the chlorophyll content of plants by interfering with chlorophyll biosynthesis. Another talk on the physiological effects of anthropogenic freshwater pollutants on duckweeds was presented by Darlielva do Rosario Freitas (Federal University of Viçosa, Brazil). She discussed the significance of the chemical form of iron supply in alleviating cadmium toxicity to *Lemna valdiviana*. Iron modulated the antioxidant system of duckweed plants depending on whether it was applied as zerovalent nanoparticles or in ionic form and also affected plant responses to cadmium treatments. With regard to duckweed stress responses, Viktor Olah (University of Debrecen, Hungary) approached this topic from a different aspect by analyzing spatial patterns in photosynthetic responses, starch and anthocyanin content at the level of individual fronds. His results indicated that there were considerable differences in duckweed responses depending on frond ontogeny, species and stressor applied.

By what mechanisms can clonal plants improve their stress resistance? One possible way to achieve higher diversity can be polyploidization [5]. Quinten Bafort (Ghent University, Belgium) gave insights in his invited talk to the phenotypic responses of tetraploid *S. polyrhiza* strains to gradients of environmental factors and their implications on fitness of the neopolyploid populations [21]. One other way to cope with stressors in monoclonal populations can be the inheritance of non-genetic phenotypic traits, such as through epigenetic mechanisms or incorporation of a microbiota [22]. Another invited talk by Alexandra Chávez (University of Muenster, Germany) addressed this topic by analyzing responses of many *S. polyrhiza* genotypes under biotic and abiotic stress for several frond generations. It was concluded that both formation of non-genetic based phenotypes as well as selection of these phenotypic variants improved stress resistance of clonal *S. polyrhiza* populations, and this effect can last for multiple generations even after stress release.

Whether focusing on basic research topics or practical utilization, many duckweed applications rely on the diversity of Lemnaceae. Thus, maintaining and further expanding duckweed stock collections with naturally evolved or artificially created clones is a necessity of the job [23]. Maintenance of large duckweed collections may, however, become very demanding in terms of infrastructure and human resource. Two presentations by Shogo Ito (Kyoto University, Japan) and by Manuela Nagel (Leibniz Institute of Plant Genetics and Crop Plant Research, Germany) discussed the potentials and pitfalls of long-term cryopreservation of duckweed. This method, besides its promise of a more cost-efficient way to maintain large-scale duckweed stock collections, can also decrease the risk of mixing stocks, or inadvertent infections of the cultures with microbes. Both groups tested a wide range of cryopreservation protocols and parameter settings to achieve successful preservation and regeneration of duckweed plantlets after their storage under cryogenic conditions. The presented results suggest that long-term cryopreservation of duckweed strains is achievable, but species-specific protocols may be needed, while some clones from different geographical latitudes showed different success rates in regeneration after cold storage.

## 6. Wastewater Remediation

Duckweeds are the fastest growing flowering plants [24], and as aquatic plants, they are especially suitable for wastewater treatment. The importance of these applications was emphasized in a contribution via zoom link about the “New Circular Economy” (Paul Skillicorn, Skillicorn Technologies, Austin, TX, USA) that described application of duckweeds in the universe of volatile wastes produced by humans and farm animals, industrial volatile wastes and farm-crop residuals, as well as the waters that accompany them. The plants’ potential roles in cleaning up the global environment and helping to remove carbon from the atmosphere, solving global malnutrition, providing rural jobs and bestowing wealth on smallholder farmers, were explored. A research group from Cork, Ireland, also developed an “Integrated MultiTrophic Aquaculture”-system (IMTA) and applied it for dairy wastewater remediation. A new cascading system for valorization of dairy wastewater couples microbial-based technologies of anaerobic digestion and/or aerobic dynamic feeding with duckweed cultivation. As part of the indoor system, duckweed is grown in a stacked flow-through system to minimize the spatial footprint. This system is not only useful for cleaning large volumes of dairy wastewater, it produces also large amount of biomass all year round (Marcel Jansen, University College Cork, Ireland; see [25]). In another report, duckweed-based IMTA was used to test the removal of total nitrogen and phosphorus from wastewater. This system united large-scale wastewater purification with production of biomass that was used for fish feeding (rainbow trout and European perch) (Simona Paolacci, Bantry Marine Research station, Ireland, see [26]). Both reports from Ireland represent a milestone in the practical application of duckweeds.

## 7. Large Scale Cultivation

The required up-scaling from lab-scale cultivation of duckweeds in the range of some grams of fresh weight, to production in the range of tonnes of biomass being generated, is presently a bottleneck for duckweed applications. Some successful developments were already described in the previous section (Paul Skillicorn, USA; Marcel Jansen, Ireland; Simona Paolacci, Ireland). Under the brand name “LemnaCore 1.0”, a group from Germany presented a mobile duckweed cultivation system, which is based on recycling and utilization of industrial process and wastewater. A vertical aquafarming technology was devised that cultivates biomass on connected levels to reduce the footprint required. The goal is to create an environment for duckweed in which it can grow well, and the production is reproducible and automated (Marko Dietz, Carbon Clouds, Germany). In another project, the development of a large-scale, vertically-integrated duckweed production system was started with optimizing abiotic factors (nutrient medium, light conditions) for the cultivation of *L. minor* 9441 and *W. hyalina* 9525 from a small culture. Goals of this optimization were not only high growth rates and biomass with high protein content but also prevention of competing growth of algae [27,28]. On the basis of these results, a recirculating, large indoor vertical farm for duckweed biomass production was developed [29] possessing nine levels for cultivation. The water is re-circulating within the system, while nutrients, temperature and light settings are controlled and regulated automatically. Per day more than 900 g of fresh biomass is harvested. Further up-scaling and improved levels of regulation are in preparation (Finn Petersen, University for Applied Sciences Osnabrück, Germany).

## 8. Feed and Food

Another highlight of the 6th ICDRA was the report that duckweed for human nutrition could be cultivated under sterile conditions and in large scale, using lighted cabinets that are sealed off from the outside environment. The company GreenOnyx, Israel, introduced a breakthrough farming technology to grow and deliver duckweed vegetables, called Wanna greens^®^, to the global market. Whereas any duckweed species can theoretically be used for this technology, the focus at the moment is on two species from the genus *Wolffia*, *W. globosa* and *W. arrhiza*. Based on an array of compact-modular growing systems that are sealed and sterile, a fully automated supply chain was introduced (Tsipi Shoham, GreenOnyx, Israel). It should be mentioned that this company obtained the permission from the European Commission to use and sell fresh plants of *W. globosa* and *W. arrhiza* as “traditional food”, and not considered as “novel food” [30]. Investigations to obtain the same permission for *Lemna* species (*L. gibba*, *L. minor*) are on the way (Ingrid van der Meer, Wageningen University, Netherlands; see [31]). Studies on the nutritional value and digestibility of duckweed protein were carried out and provided promising results (Jurriaan Mes, Wageningen Food and Biobased Research, Netherlands; see [32]).

Known for a long time, duckweeds can also be used for animal feeding. The nutritional value of duckweed as feed has already been confirmed in several studies with broiler chickens. However, contrasting effects of using duckweed in complete diets on growth and especially feed intake have been found. The digestibility of crude protein can differ strongly along with the quality of duckweed batches, ranging from 68 to 90% for methionine. The differences are most probably caused by higher contents of tannins and fiber. Different cultivation conditions for duckweeds are being tested further to evaluate the factors more critically (Johannes Demann, University for Applied Sciences, Osnabrück, Germany).

## 9. Duckweed Futures: Visions of Things to Come

It is fitting in closing this meeting summary to reflect on the advances in duckweed research and applications that have been made over the past eleven years since the first ICDRA in Chengdu, China [2]. At that time, the first draft genome of *S. polyrhiza* had just been completed to the sequencing phase by the JGI (Joint Genome Institute, USA) using a combination of 454 and Illumina (San Diego, CA, USA) sequencing technologies. In those days, BAC-end sequencing of large genomic fragments was being carried out to use as scaffolding tools for genome assembly [33]. Today, we have in hand reference quality genomes for all five genera of duckweeds and for species such as *S. polyrhiza*, multiple large-scale population genomic datasets with hundreds of accessions are available to all. This fast-forward genomics approach would likely accelerate even more over the next decade due to rapid improvements in long-read, single-molecule sequencing technologies, notably Oxford Nanopore and PacBio being orthogonal approaches that are increasing their fidelity while decreasing time and costs. In addition, further diversification of these technologies to rapidly quantifying epigenetic marks on DNA can be used to broaden their impact in biology as well. The high quality reference genome collection will likely be completed for all 36 known species within the next decade, if not sooner. Their availability will enable researchers to begin a quantitative understanding of how this family has evolved over the millions of years since its divergence from the early monocot ancestor. Furthermore, their comparative analysis to other terrestrial plants should open new windows into alternative strategies for these miniature plants to adapt to vastly diverse environments. A case in point is the discovery of the severe loss of most of the immune-related NLR genes in *W. australiana* while this species has elaborated more genes in the PRR-dependent basal immunity pathways [18]. The advance in genomics will likely drive new discoveries in the role of epigenetic pathways in duckweed biology. As the comparison of turion and frond DNA methylation in *S. polyrhiza* has shown [6], there are clearly significant differences in the way DNA are being methylated during this developmental transition, even in a duckweed species that has the lowest overall cytosine methylation in plants studied so far [34]. Future examination of this behavior in other species of duckweeds and response pathways will likely open many new avenues for epigenetic studies in duckweed, as well as providing new understanding of how things can be performed differently for similar outcomes.

The areas of duckweed farming and product development will likely forge close relationships with many types of basic research interests in ecology, evolution and chemical genomics in the coming years. One approach that has gained popularity in the past few years is the interest in modular indoor growth system for duckweed, in contrast to the more traditional usage of raceway ponds and lagoons that have been deployed in earlier academic research and by companies such as Parabel USA Inc. In this ICDRA, there are at least five groups who have presented talks or posters involving vertically integrated duckweed growth systems for scalable indoor production, which is a sign that this approach is being adopted and developed by our community at large. While the indoor hydroponic growth system for plant production is not novel per se, the parameters and precise constraints posed by an aquatic plant such as duckweed are new and unique. The automation and plant maintenance knowledge for optimized duckweed growth in this scenario would need to be established for these new experimental systems, and their availability down the line as user-friendly growth systems will not only open new opportunities for cropping duckweed, but also will provide useful platforms for basic research under controlled environmental conditions. For example, this could help accelerate phenotyping approaches to identify important traits and test for effects by chemicals or biological agents such as beneficial microbes on duckweed health and productivity.

Finally, there are two bottlenecks for duckweed research and applications that we see as potential breakthroughs in the coming years that would help to further transform the community. One is the functional understanding of flowering control in duckweed that will enable effective and routine genetic manipulation through crossing and selection for varieties of interest. This capability could be facilitated by a molecular understanding of the basis for key flowering inducing genes and compounds. Its success would usher in an era of duckweed breeding via molecular genetics. The second hurdle that we would like to see overcame is the optimization and deployment of a robust and rapid in planta transformation technology for duckweeds. This ability will enable rapid molecular studies for genes of interest that have been defined from a rapidly growing database of transcriptomics data being generated by our community. A reliable and facile method that can overcome the often-observed strain and species dependence of transformation protocols [35] to investigate gene functions and pathways will truly open the door to the fascinating world of duckweeds in the coming decade.

## Figures and Tables

**Figure 1 plants-12-02134-f001:**
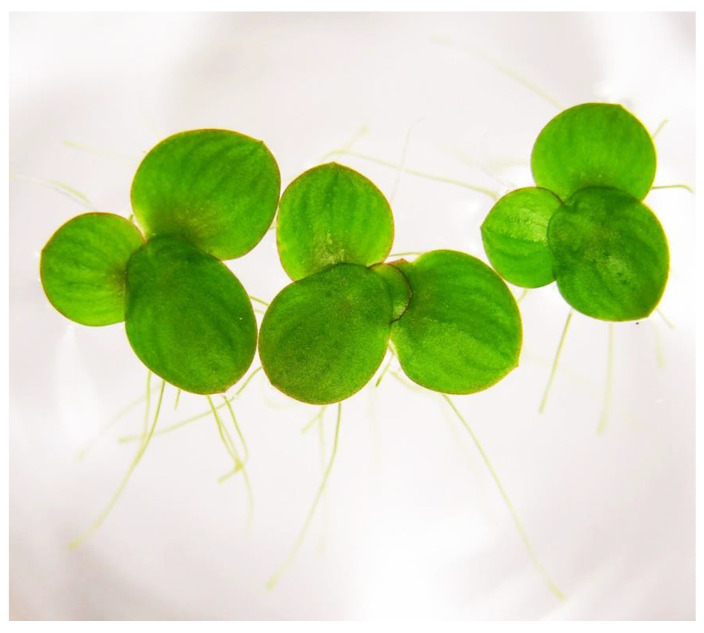
Colonies of *Spirodela polyrhiza* showing the frond architecture.

**Figure 2 plants-12-02134-f002:**
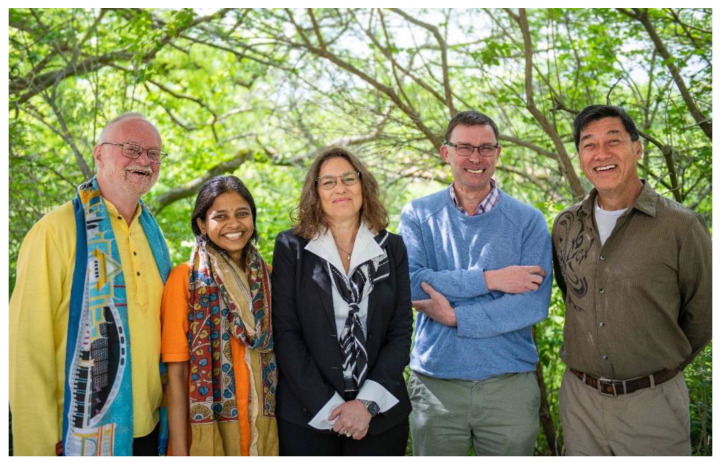
International Steering Committee on Duckweed Research and Applications (ISCDRA). From left to right: Klaus-J. Appenroth (University of Jena, Germany), K. Sowjanya Sree (Central University of Kerala, India), Tsipi Shoham (GreenOnyx, Tel Aviv, Israel), Marcel Jansen (Cork University, Ireland), Eric Lam (Rutgers University, New Brunswick, NJ, USA; Chair of the committee). (Photo courtesy: IPK Archives).

**Figure 3 plants-12-02134-f003:**
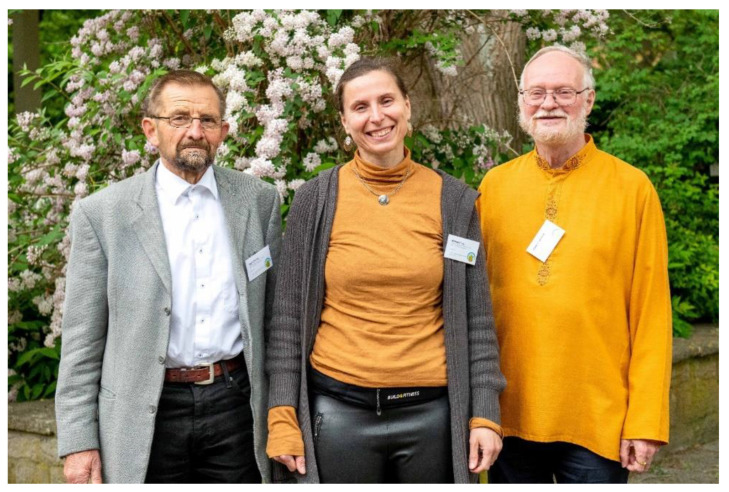
Scientific organization committee of the 6th ICDRA. From left to right: Ingo Schubert (IPK Gatersleben, Germany; Chair of the conference), Manuela Nagel (IPK Gatersleben, Germany) and Klaus-J. Appenroth (University of Jena, Germany), Co-Chairs. (Photo courtesy: IPK Archives).

**Figure 4 plants-12-02134-f004:**
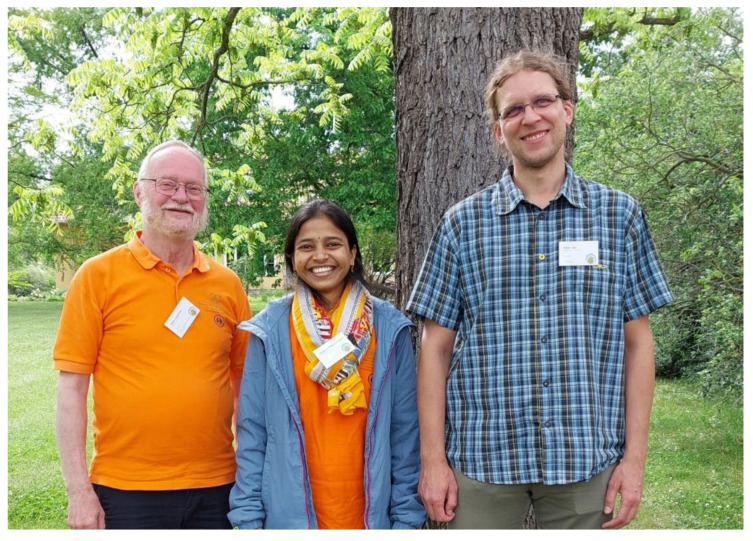
Guest editors of the Special Issue “Duckweeds: Research meets applications” of the journal *Plants*. From left to right: Klaus-J. Appenroth (University of Jena, Germany), K. Sowjanya Sree (Central University of Kerala, India), Viktor Oláh (University of Debrecen, Hungary). (Photo courtesy: IPK Archives).

## Data Availability

All the data related to this publication are mentioned in the text.

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
