# Peer review of "Sixth International Conference on Duckweed Research and Applications Presents Lemnaceae as a Model Plant System in the Genomics and Postgenomics Era"

_plants, 2023, doi:10.3390/plants12112134_

Round 1
Reviewer 1 Report
The manuscript represents a comprehensive record of the recent biannual conference on Duckweed Research and Applications. The conference review is well written, informative and well structures. Except a couple of minor changes, not much improvements could be made:
Page 4: Structure of a sentence starting with “Moreover, that triploid hybrids are commonly found that have arose from Lemna minor…” could be improved.
Page 4 and 5: I would recommend to be consistent with addressing the presenters by using their first and last names, like: Nikolai Borisjuk, Alexander Ware, Anton Stepanenko.
Author Response
Dear Reviewer,
Many thanks for the suggestions. We have incorporated all the changes as suggested. We are uploading a track change version for conveniently locating the changes.
Sincerely,
Dr. K. Sowjanya Sree
Reviewer 2 Report
The manuscript by Olah et al. reports on the 6th International Conference on Duckweed Research and Applications (6th ICDRA) in 2022 in Germany. The Conference Report provides extensive coverage of the key topics presented during the conference, showcasing the latest advancements in various fields of duckweed research and practical applications. The Conference Report thoroughly covers all the main topics presented at the conference reflecting the recent advances in the diverse areas of duckweed research and applications.
The review contains valuable information for a wide range of biologists and is particularly useful for those working with Duckweed. The seven thematic areas that define global Duckweed research are accurately presented in the Report: 1) Genomics, evolution and genome organization; 2) Differentiation and diversity; 3) Physiology, metabolism and microbiome; 4) Stress, toxicology, cryopreservation; 5) Wastewater remediation; 6) Large scale cultivation; and 7) Feed and food. Each theme is presented in a way that inspires specialists and raises additional critical questions, thereby advancing research in the field.
The 6th International Conference on Duckweed Research and Applications 2022 has been an important event in the history of the Duckweed Society and has been presented here in a very professional, versatile and critical way. I would welcome the acceptance of this manuscript for publication in PLANTS.
Author Response
Dear Reviewer,
Many thanks for your commnets. We appreciate it.
Sincerely,
Dr. K. Sowjanya Sree